# Kidney Replacement Treatment in South-Western Italy (Campania): Population-Based Study on Gender and Residence Inequalities in Health Care Access

**DOI:** 10.3390/jcm10030449

**Published:** 2021-01-24

**Authors:** Massimo Cirillo, Raffaele Palladino, Carolina Ciacci, Lidia Atripaldi, Maria Grazia Fumo, Roberta Giordana, Maria Triassi

**Affiliations:** 1Department of Public Health, University “Federico II” of Naples, 80131 Naples, Italy; massimo.cirillo@unina.it (M.C.); triassi@unina.it (M.T.); 2Interdepartmental Center for Research in Healthcare Management and Innovation in Healthcare (CIRMIS), University “Federico II” of Naples, 80131 Naples, Italy; 3Department of Primary Care and Public Health, School of Public Health, Imperial College of London, London W6 8RP, UK; 4Department ‘Scuola Medica Salernitana’, University of Salerno, 84081 Baronissi, Italy; cciacci@unisa.it; 5Clinical Biochemistry Unit, AORN Ospedale dei Colli, 80131 Naples, Italy; lidia.atripaldi@virgilio.it; 6Regional Healthcare Society (So.Re.Sa), 80143 Naples, Italy; m.fumo@soresa.it (M.G.F.); r.giordana@soresa.it (R.G.)

**Keywords:** kidney replacement treatment, dialysis, epidemiology

## Abstract

The aim of this study was to investigate the epidemiology of kidney replacement treatment (KRT) in Italy with a focus on gender and residence. As a population-based study using administrative databases from the Campania region of Italy between 2015 and 2018, the study outcomes included diagnoses of haemodialysis, peritoneal dialysis, kidney transplant, and mortality. A total of 11,713 residents in Campania were on KRT from 2015 to 2018. The annual prevalence ranged between 1000 and 1015 patients per million population (pmp) for haemodialysis, between 115 and 133 pmp for peritoneal dialysis, and between 2081 and 2245 pmp for kidney transplant. The annual incidence ranged between 160 and 185 pmp for de novo haemodialysis and between 59 and 191 pmp for kidney transplant. Annual mortality ranged between 12.8% and 14.2% in haemodialysis, between 5.2% and 13.8% in peritoneal dialysis, and between 2.4% and 3.3% in kidney transplant. In Cox regression targeting mortality, significant HRs were found for age (95%CI = 1.05/1.05), kidney transplant (compared to haemodialysis: 0.37/0.47), residence in suburban areas (1.03/1.24), and de novo dialysis incidence in years 2015–2018 (1.01/1.17). The annual rate of kidney transplant was 2.6%. In regression targeting kidney transplant rate, significant HRs were found for female gender (0.67/0.92), age (0.93/0.94), residence in suburban areas (0.65/0.98), and de novo incidence of dialysis in 2015–2018 (0.49/0.71). The existence of socioeconomic inequities in KRT is suggested by the evidence that gender and suburban residence predicted mortality and/or access to kidney transplant.

## 1. Introduction

The prevalence of kidney replacement treatment (KRT) is increasing worldwide in parallel with the prevalence of end-stage kidney disease [1,2,3]. KRT management has a strong economic and social impact given that morbidity and mortality remain high in patients on KRT [1,2,3]. Epidemiological data on KRT and on its outcomes are fundamental in organizing appropriate pathways for KRT management. Registries of KRT patients have been created to address this need in most industrialized countries [4,5]. In Italy, the KRT Registry has been managed by the Italian Society of Nephrology (*Società Italiana di Nefrologia*) for the past 25 years [6]. Data collection for this Registry is based on yearly reports from regional sections of the society and is still largely incomplete because only a minority of sections report complete data [7]. Administrative databases have been successfully used to build up registries for various diseases including chronic kidney disease (CKD) [8,9]. These methods have been applied also to Italian regional administrative databases [10,11]. The present study analysed administrative databases of the Campania region, southern Italy, to investigate the epidemiology of KRT and, in particular, the possibility that gender and residence could be determinants of health care inequality in the field of kidney diseases [12,13,14,15,16].

## 2. Materials and Methods

The KRT provided by the Italian national health system includes haemodialysis, peritoneal dialysis, and kidney transplant [6]. Temporary dialysis due to acute kidney failure, either haemodialysis or peritoneal dialysis, is provided as in-patient treatment by public or private hospitals. Chronic haemodialysis can be provided as out-patient treatment by hospitals and/or private ambulatory specialty clinics accredited by the health authority and part of state-run health care. Chronic peritoneal dialysis and follow-up of patients with kidney transplants can be provided by public or private hospitals as day-surgery or hospitalization depending on the clinical conditions.

The Italian national health system is characterized by separate regional administrative authorities that are further structured in local health authorities. Seven are the local health authorities of the Campania region: one for the city of Naples (Napoli 1), two for the suburban areas around the town of Naples (Napoli 2 nord and Napoli 3 sud, respectively), and four for each one of remaining provinces (Avellino, Benevento, Caserta, and Salerno). The present study collected Campania region administrative data for all forms of KRT including haemodialysis, peritoneal dialysis, and kidney transplant. Data collection excluded non-residents in Campania. The analysis was based on the data extracted from several administrative databases for the period 2014–2018, which included hospital discharge records, outpatient specialty visits, and drug prescriptions. In addition, the database of the local civil registry for the same study period was accessed to collect mortality data. All data were linked and fully anonymized. The retained information included KRT-specific information together with sex, date of birth, date of entry in the database, date of death in the case of death, and health authority of residence.

### 2.1. Prevalence and Incidence

The prevalence of KRT and its specific subtypes was defined for years 2015–2018 as follows:-haemodialysis, if data of the hospital discharge records or outpatient nephrology records included code V560 or codes 39951–59 (ICD-IX CM);-peritoneal dialysis, if data of the hospital discharge records included code V451 or codes 54981–82 (ICD-IX CM) and/or if data of the drug prescription records included any of F00001678-73, LPB5262G-64G, LPB96564, LPB5268, or LCE8280-81 (Italian national formulary);-kidney transplant, if data of the hospital discharge records included codes V420 and/or 99681 (ICD-IX CM) and/or if data of the outpatient nephrology records included code 897 (ICD-IX CM).-A person was defined on dialysis treatment if he/she was on treatment for haemodialysis and/or peritoneal dialysis. In the case of multiple treatments, priority was given to the treatment with lower frequency. Therefore, priority was given to peritoneal dialysis when haemodialysis and peritoneal dialysis were prevalent or incident in the same year and was given to kidney transplant when dialysis and kidney transplant were prevalent or incident in the same year. Prevalence data of the year 2014 were used only for the definition of KRT incidence in 2015 and were not reported. The de novo incidence of KRT and the type of incident KRT were defined for years 2015–2018 if data of that year included the codes listed above for prevalence and if the patients were without prevalent KRT in the previous year. As per the definition, the de novo incidence of KRT did not include patients receiving kidney transplant after a period of dialysis and/or patients reinitiating dialysis after kidney transplant failure.

### 2.2. Outcomes

For each year of prevalent KRTs, the following outcomes were defined: hospitalization within the same year; death within the same year; and for surviving patients, type of KRT in the subsequent year. For any given year, prevalent patients were defined as missing to datafile (“*missing*”) if the patient did not appear in the KRT database of the subsequent year or in the mortality database. The type of prevalent KRT in the subsequent year and missing patients could not be defined for prevalent patients of year 2018 because data collection did not include databases for year 2019.

### 2.3. Statistics

Prevalence and incidence were reported for years 2015–2018 as absolute counts (N), as percent (%), or as counts of patients per million population (pmp). Counts of residents in Campania per year were derived from the Italian National Institute of Statistics [17]. The results as pmp for the whole region included 95% confidence interval (95%CI) for comparisons with other studies. Priority in graphic presentation was given to results of prevalence and incidence for the year 2017 to allow comparisons with detailed data reported for the same year reported in a subgroup of Italian regions not including Campania (7). Graphic presentations of prevalence and incidence for the other years are included in the Appendix A (years 2015–2016 and 2018).

Correlates of mortality and access to kidney transplant were analysed using Cox proportional hazard regression models. Analyses on mortality targeted all patients, whereas analyses on kidney transplant targeted only dialysis patients, excluding patients with prevalent kidney transplant since the first entry in the database. For analyses on mortality, the observation period was calculated as the difference from 1 January 2015 or the date of entry into the database if later to the date of death or to 31 December 2018 for surviving patients. For analyses on kidney transplant, the observation period was calculated as the difference from 1 January 2015 or the date of entry in database if later to the date of kidney transplant or to 31 December 2018 for patients not receiving kidney transplant. Independent variables included in the regression were gender, age at entry in datafile, type of KRT, local health authority of residence, and de novo incidence of dialysis in 2015–2018 to control also for possible differences between KRT lasting ≤ years and KRT lasting > 4 years. The two local authorities of the Naples suburban areas were combined together in analysis for comparison to the urban area of the city of Naples. An assumption of hazard proportionality for the Cox models was checked both graphically and statistically. The results of the Cox analyses were reported as hazard ratio (HR) with 95%CI. Statistical procedures were performed using Stata 15 MP and/or SPSS-19.

## 3. Results

### 3.1. Prevalence and Incidence

A total of 11,713 residents in Campania with KRT from 2015 to 2018 were identified. The majority of patients were male, with age ≥ 45 years, and residents in the urban or suburban areas of Naples (Appendix A).

The upper section of Table 1 shows that, from 2015 to 2018, the prevalence ranged (min–max) from 1375 to 1414 pmp for KRT, from 1000 to 1015 pmp for haemodialysis, from 115 to 133 pmp for peritoneal dialysis, and from 2081 to 2245 pmp for kidney transplant. Prevalence as pmp increased every year up to 2018 only for KRT and kidney transplant. The overall prevalence from 2015 to 2018 was a total of 8236 patients for haemodialysis, 237 patients for peritoneal dialysis, and 3240 patients for kidney transplant. In all years, more than 98% of prevalent haemodialysis patients were treated in private outpatient clinics.

The lower section of Table 1 shows that, from 2015 to 2018, the annual incidence ranged from 166 to 192 pmp for de novo KRT, from 160 to 185 for de novo haemodialysis, from 29 to 53 pmp for de novo peritoneal dialysis, and from 159 to 191 pmp for kidney transplant. De novo incidence from 2015 to 2018 as pmp varied without a consistent trend over time for all types of KRT.

### 3.2. Associations with Gender and Age

Prevalence increased with age in both sexes for KRT, haemodialysis, and kidney transplant (Figure 1 and Appendix A). Peak prevalence was found at ages ≥ 75 years for KRT and haemodialysis and at ages 55–64 for kidney transplant. Prevalence was <100 pmp at all ages for peritoneal dialysis, which was the most prevalent dialysis treatment only for ages < 18 years (Appendix A). Peak prevalence of KRT and haemodialysis shifted toward older ages in males from 2015 and in females in 2018 (Appendix A). For all ages ≥ 35 years, haemodialysis and kidney transplant were apparently more prevalent in men compared to women.

The de novo incidence of KRT and haemodialysis increased with age and peaked at ages ≥ 75 years in males and females (Figure 2 and Appendix A). The incidence of peritoneal dialysis was <50 pmp for all ages in both sexes. The incidence of kidney transplant increased with age, peaking at ages 45–54 in men and at ages 55–64 in women. The de novo incidence of KRT and haemodialysis was apparently higher in men for all ages ≥ 35 years. These trends were consistent in other years although the peak prevalence of KRT and haemodialysis as pmp shifted toward older ages (Appendix A). The incidence of kidney transplant was inversely associated with age in both genders (Appendix A).

### 3.3. Outcomes

Figure 3 summarizes the results of the outcomes for prevalent patients on haemodialysis, on peritoneal dialysis, and with kidney transplant separately. The upper panels show the results for death or hospitalization within the same year, whereas the lower panels show results in the subsequent year for type of KRT and missing data. Shown as the percent of patients on treatment, death within the year ranged between 12.8% and 14.2% for haemodialysis, between 5.2% and 13.8% for peritoneal dialysis, and between 2.4% and 3.3% for kidney transplant. Hospitalization within the year ranged between 35.2% and 36.5% for haemodialysis but was stably 100% for peritoneal dialysis and kidney transplant. For patients on haemodialysis, the prevalence of KRT in the subsequent year ranged above 78.8% for haemodialysis, below 0.3% for peritoneal dialysis, and between 2.0% and 3.8% for kidney transplant. For patients on peritoneal dialysis, the prevalence of KRT in the subsequent year ranged above 48.8% for peritoneal dialysis, below 11.4% for haemodialysis, and between 7.5% and 12.0% for kidney transplant. For patients with functioning kidney transplant, the prevalence of KRT in the subsequent year ranged above 82.0% for kidney transplant, below 4.3% for haemodialysis, and below 0.2% for peritoneal dialysis. The prevalence of missing KRT data in the subsequent year ranged between 2.0% and 3.8% for patients on haemodialysis, between 10.4% and 15.8% for patients on peritoneal dialysis, and between 9.9% and 11.0% for patients with functioning kidney transplant.

### 3.4. Correlates of Mortality and Kidney Transplant

The annual mortality rate in the 4-year period from 2015 to 2018 was 11.7% for KRT (31221 patient-years), 16.5% for haemodialysis (19669 patient-years), 11.7% for peritoneal dialysis (624 patient-years), and 3.2% for kidney transplant (10927 patient-years). The difference among KRT types was highly significant (log rank chi-square 1063, *p* < 0.001). Table 2 summarizes the results of the Cox regression. Compared with haemodialysis, kidney transplant was associated with a 59% reduction in all-cause mortality (*p* < 0.001) whereas independent correlates of higher mortality were older age at entry in database (*p* < 0.001), residence in the suburban areas of Naples and in the Avellino province compared to residence in the town of Naples (*p* = 0.007 and 0.044, respectively), and de novo incidence of dialysis in years 2015–2018 (*p* = 0.0.037). When the analyses were separately re-run by type of KRT, the findings for kidney transplant patients indicated an association of female gender with lower mortality rate (*p* < 0.041, Appendix A).

In the 4-year period from 2015 to 2018, annual rate of kidney transplant was 2.6% (21,234 patient-years). Cox regression indicated that the independent correlates for lower rate of kidney transplant were female gender (*p* = 0.002), older age at entry in database (*p* < 0.001), residence in the suburban area of Naples compared to residence in the town of Naples (*p* = 0.033), and de novo incidence of dialysis in years 2015–2018 (*p* < 0.001). HR of kidney transplant was not significant for type of dialysis and residence in other provinces (*p* > 0.06).

## 4. Discussion

This study reports the first analysis on the epidemiology of KRT in Campania, a region for which data were traditionally incomplete or missing in the reports of the Italian and European registries of KRT. The present results for years 2015–2018 indicated a prevalence of around 1300–1400 patients pmp and an incidence of around 160–190 patients pmp, that is, rates in the intermediate range of the European registry [5] and in accordance with reports from other Italian regions [7,10]. The most common type of KRT was by far haemodialysis in private ambulatory specialty clinics. Peritoneal dialysis was the least prevalent dialysis in adults but was the most prevalent dialysis in paediatric ages. The annual mortality rate among KRT patients was about 11%, in accordance with reports from other Italian regions [7,10,18]. Mortality rate associated independently with age, type of KRT, residence, and recent de novo incidence of dialysis. The annual rate of kidney transplant ranged was about 3% and was associated with gender, age, residence, and recent de novo incidence of dialysis.

The main limitations of this study are the lack of information on the cause of end-stage kidney disease, comorbidities, and KRT initiation date, a datum that the per study design was lacking for all patients on KRT for five or more years. The lack of information on ethnicity is a minor limitation because the percent prevalence of foreign individuals residing in Italy totals less than 9%, including European and non-European individuals [17].

Two findings supported the interpretation that chronic dialysis represented at least 93% of cases defined on dialysis treatment in the present study. First, haemodialysis accounted for more than 95% of KRT other than kidney transplant; second, over 98% of haemodialysis patients were treated in private ambulatory specialty clinics where acute or temporary haemodialysis cannot be performed. Cases defined as “missing”—that is, cases absent in the databases of mortality and KRT of the subsequent year—may represent a combination of three different subgroups: temporary dialysis due to acute kidney failure, change of residence from Campania to another region, and inappropriate coding. In comparison to haemodialysis patients, the percent of missing cases was 2–3 times higher among patients on peritoneal dialysis and patients with kidney transplant. This suggested that inappropriate coding was much rarer in private ambulatory specialty clinics that depend on accurate certifications to receive pay back from the regional health authority.

The results about KRT in the subsequent year indicated that haemodialysis was the most common option also for patients requiring dialysis after kidney transplant failure, consistent with present data for prevalence and incidence and with reports from a region of north-eastern Italy [18]. Likewise, the switch from peritoneal dialysis to haemodialysis was much more common than the opposite one.

The year-to-year variability in prevalence and incidence supports the idea that a variability over time contributes to explaining part of the inter-region variability. The change in KRT prevalence from 2015 to 2018 was slightly higher as pmp than as absolute counts (+2.8% and +1.3%, respectively) because the Campania population decreased from 2015 to 2018 (footer of Table 1). Data of the Italian Dialysis and Transplantation Registry indicated an increasing trend in KRT incidence up to 2011, followed by a stabilization of the incidence [7]. The present results indicate that the temporal trends could differ among different regions. The higher KRT prevalence in Campania was due exclusively to haemodialysis in persons aged ≥45 years, given that, compared to data of the Italian Dialysis and Transplantation Registry, the curve of prevalence by age was almost identical for kidney transplant and actually lower for peritoneal dialysis (Appendix A).

Mortality rate was independently higher with increasing age in dialysis patients compared to patients with kidney transplant and in patients with de novo incidence of dialysis, in accordance with previous reports [18,19]. Considering that female gender associates with lower mortality rate also in the presence of severe stages of chronic kidney disease [20] and that, in the present study, gender associated with mortality in kidney transplant but did not associate with mortality in dialysis patients, the available data suggest that dialysis implies the loss of the advantage of a lower mortality risk typical of female gender in the presence of reduced kidney function. The evidence of a consistent association of suburban residence with a higher mortality rate among patients on dialysis indicated an effect of socioeconomic factors on the outcome of this treatment and the existence of inequalities between urban and suburban areas [12]. To our knowledge, the role of socioeconomic factors in mortality of dialysis patients was previously investigated only in non-European countries [21]. The possibility that socioeconomic factors affect comorbidities and KRT initiation could not be investigated due to the lack of information on these variables.

Female gender, older age, suburban residence, and de novo incidence of dialysis were all independently associated with less frequent access to kidney transplant. The present results for the associations with older age and KRT initiation date were in accordance with the data of large registries [4,5]. To our knowledge, an association of female gender with less frequent kidney transplant was not previously reported in Europe or Italy but was described in the US and Canada [13,15]. The association could be an effect of several factors including inequalities in men and women enrolment in the transplantation list, a higher prevalence of contraindications in kidney transplant in women, and a gender bias in the transplant centres. The association of suburban residence with rarer access to kidney transplant reinforced the idea of social inequities in KRT because it paralleled the association of suburban residence with a higher mortality rate.

The practical implications of this study are linked to the possibility to improve the planning and organization of medical and social assistance to KRT patients on the basis of objective and reliable information about prevalence, incidence, and outcomes. The novel findings, at least for Italy and Europe, of a higher mortality rate and rarer access to kidney transplant in dialysis patients with suburban residence highlights the possibility of significant social inequities in KRT. The possibility is further supported by the observation of rarer access to kidney transplant for females as well.

## 5. Conclusions

In summary, this study reported the first analysis of the epidemiology of KRT in the third most populous Italian region situated in south-western Italy. Prevalence and incidence were in the intermediate ranges reported by the European registry. In KRT patients, age, KRT initiation date, and KRT type were independent predictors of mortality and access to kidney transplant. The existence of socioeconomic inequities was suggested by the evidence that gender and suburban residence are predictors of mortality and access to kidney transplant.

## Figures and Tables

**Figure 1 jcm-10-00449-f001:**
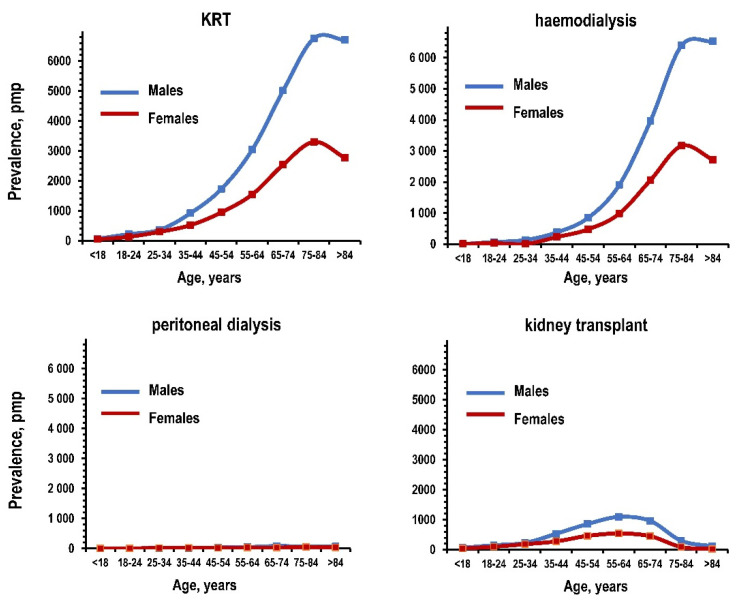
Prevalence of kidney replacement treatment, haemodialysis, peritoneal dialysis, and kidney transplant among residents of the Campania region in the year 2017: patients per million population (pmp) by age and gender. Notes: blue line for males and red line for females. KRT: kidney replacement treatment.

**Figure 2 jcm-10-00449-f002:**
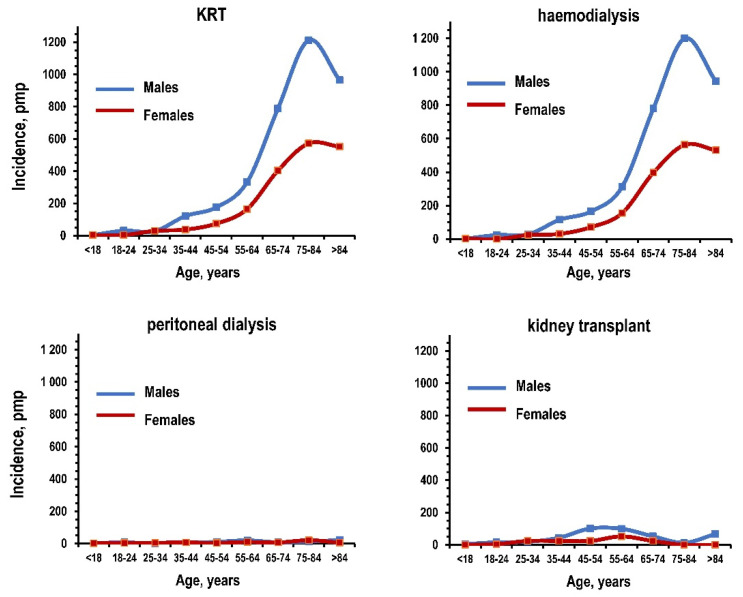
De novo incidence of kidney replacement treatment, haemodialysis, peritoneal dialysis, and kidney transplant among residents of the Campania region in the year 2017: patients per million population (pmp) by age and gender. Notes: blue line for males and red line for females. KRT: kidney replacement treatment.

**Figure 3 jcm-10-00449-f003:**
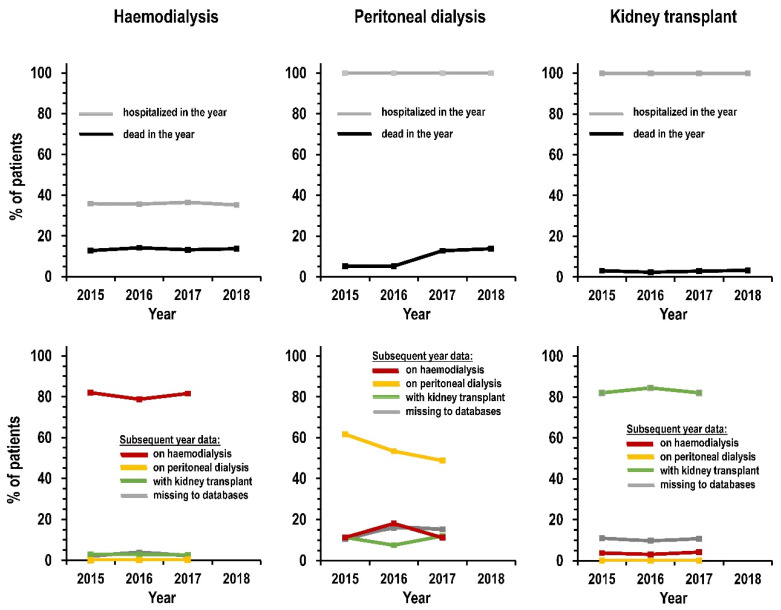
Prevalence (%) by type of kidney replacement treatment and by calendar year. Notes: graphs display patients with at least one hospitalization within the year (upper panels, grey line); dead within the year (upper panels, black line); on haemodialysis in the subsequent year (lower panels, red line); on peritoneal dialysis in the subsequent year (lower panels, yellow line); not on dialysis and with kidney transplant in the subsequent year (lower panels, green line); and missing from kidney replacement treatment and mortality databases of the subsequent year (lower panels, grey line).

**Table 1 jcm-10-00449-t001:** Prevalence and de novo incidence of kidney replacement treatment among residents of Campania by calendar year: absolute counts (N) and count per million population (pmp) with 95%CI.

		Calendar Year			
		2015	2016	2017	2018
**Prevalence**					
KRT	N pmp (95%CI)	8057 1375 (1302–1447)	8173 1397 (1324–1470)	8218 1407 (1334–1481)	8167 1414 (1340–1487)
Haemodialysis	N pmp (95%CI)	5861 1000 (938–1062)	5939 1015 (953–1078)	5853 1002 (940–1064)	5806 1005 (943–1067)
Peritoneal dialysis	N pmp (95%CI)	115 20 (11–28)	133 23 (13–32)	125 21 (12–30)	116 20 (11–29)
Kidney transplant	N pmp (95%CI)	2081 355 (318–392)	2101 359 (322–396)	2240 384 (345–422)	2245 389 (350–427)
**Incidence**					
KRT	N pmp (95%CI)	1022 174 (148–200)	1088 186 (159–213)	1121 192 (165–219)	958 166 (141–191)
Haemodialysis	N pmp (95%CI)	993 169 (144–195)	1035 177 (151–203)	1079 185 (158–211)	922 160 (135–184)
Peritoneal dialysis	N pmp (95%CI)	29 5 (1–9)	53 9 (3–15)	42 7 (2–12)	36 6 (1–11)
Kidney transplant *	N pmp (95%CI)	159 27 (17–37)	191 33 (21–44)	187 32 (21–43)	172 30 (19–40)

Notes: Number of residents in Campania from years 2015 to 2018: 5,861,529, 5,850,850, 5,839,084, and 5,777,616. * In patients on dialysis treatment in the previous year (pre-emptive kidney transplant not included). KRT: kidney replacement treatment.

**Table 2 jcm-10-00449-t002:** Cox regression analyses on mortality and access to kidney transplant (dependent variables) among residents in the Campania region on kidney replacement treatment from 2015 to 2018: uni- and multivariate HR (95%CI), bold character for HR significantly ≠ 1.

	Dependent variable: mortality (n patients = 11,713, n events = 3666)	
Independent variables	Uni-variate HR (95%CI)	multi-variate HR (95%CI)
Female gender, y-n = 1–0	1.05 (0.98–1.12)	0.95 (0.89–1.02)
Age at entry in datafile, years	1.06 (1.06–1.06)	1.05 (1.05–1.05)
Type of KRT		
Haemodialysis	4.06 (3.61–4.56)	1 (reference)
Peritoneal dialysis, y − n = 1–0	0.80 (0.64–1.01)	0.88 (0.70–1.10)
Kidney transplant, y − n = 1–0	0.19 (0.17–0.21)	0.41 (0.37–0.47)
Local authority of residence		
City of Naples	0.94 (0.86–1.02)	1 (reference)
Suburban areas of Naples*	1.05 (0.98–1.12)	1.13 (1.03–1.24)
Avellino	1.22 (1.08–1.37)	1.15 (1.01–1.32)
Benevento	1.09 (0.93–1.27)	1.05 (0.89–1.25)
Caserta	1.02 (0.93–1.12)	1.09 (0.97–1.22)
Salerno	0.88 (0.81–0.96)	1.00 (0.90–1.11)
De novo incidence of dialysis, y − n = 1–0	1.54 (1.43–1.66)	1.08 (1.01–1.17)
	Dependent variable: kidney transplant (n patients = 8473, n events = 549)	
Independent variables	Uni-variate HR (95%CI)	multi-variate HR (95%CI)
Female gender, 1–0	0.82 (0.70–0.95)	0.79 (0.67–0.92)
Age at entry in datafile, years	0.94 (0.93–0.94)	0.94 (0.93–0.94)
Type of KRT		
Haemodialysis	0.26 (0.20–0.34)	1 (reference)
Peritoneal dialysis, y − n = 1–0	2.42 (1.84–3,19)	1.22 (0.91–1.63)
Local authority of residence		
City of Naples	1.05 (0.88–1.26)	1 (reference)
Suburban areas of Naples *	0.99 (0.85–1.15)	0.80 (0.65–0.98)
Avellino	0.91 (0.68–1.23)	0.88 (0.63–1.23)
Benevento	1.22 (0.87–1.71)	1.24 (0.86–1.79)
Caserta	1.03 (0.84–1.26)	0.83 (0.64–1.06)
Salerno	0.93 (0.76–1.12)	0.79 (0.62–1.01
De novo incidence of dialysis **, y − n = 1–0	0.65 (0.54–0.79)	0.59 (0.49–0.71)

* Suburban area of Napoli 2 nord and suburban area of Napoli 3 sud combined together, ** incidence from 2015 to 2018 included. KRT: kidney replacement treatment.

## Data Availability

Data can be obtained upon request from the Regional Healthcare Society (So.Re.Sa), Naples, Italy (www.soresa.it).

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
