# Peer review of "Kidney Replacement Treatment in South-Western Italy (Campania): Population-Based Study on Gender and Residence Inequalities in Health Care Access"

_jcm, 2021, doi:10.3390/jcm10030449_

Round 1

Reviewer 1 Report

Cirillo and co-authors present their findings on the use of kidney replacement therapy in Campania, Italy. The work is necessary due to incomplete national registry records. Overall the work is of high quality and the standardised reporting allows for international comparisons.

I have only minor comments / suggestions:

  1. I wasn't quite clear on the source of the denominator used for population estimates. I may have missed this, but what is the source of the Campania population estimates and how reliable are these?
  2. I find the figures challenging to interpret on their own. Whilst the footnotes do provide explanation, I wonder if more definitive subtitles may provide more specific information, without having to go back and forth to teh footnotes when working our gender, hospitalisation, death etc.

Reviewer 2 Report

the paper is well written and interesting

as suggested by the authors the absence of data on the cause of renal failure and comorbidities is a weakness.

my concrete suggestions would be to be explicit about the hypothesis driving the research in the introduction (not just the objectives)

in the methods explain the reasoning behing the prioritization of peritoneal/haemodialysis or transplant/dialysis

in the figures the legends are written and distracting the color legend would be better if displayed within the figure in order to get an immediate understanding and not have to look for the meaning of colors

in the tables the confidence intervals are separated by / which disturbs me (maybe because of habits) but it seems to me that usually a - is better (just a comment i am not overly dogmatic about this)

l213 close parenthesis

l 222 was is missing

l 323 i would add nuance to indicatd ...effect which implicitly brings us to causal relationships. one could argue that different neighborhoods or sexes have different prevalences of causes of end stage renal diseases or comorbidities. please rewrite a bit more carefully
